# A Descriptive-Multivariate Analysis of Community Knowledge, Confidence, and Trust in COVID-19 Clinical Trials among Healthcare Workers in Uganda

**DOI:** 10.3390/vaccines9030253

**Published:** 2021-03-12

**Authors:** Keneth Iceland Kasozi, Anne Laudisoit, Lawrence Obado Osuwat, Gaber El-Saber Batiha, Naif E. Al Omairi, Eric Aigbogun, Herbert Izo Ninsiima, Ibe Michael Usman, Lisa M. DeTora, Ewan Thomas MacLeod, Halima Nalugo, Francis P. Crawley, Barbara E. Bierer, Daniel Chans Mwandah, Charles Drago Kato, Kenedy Kiyimba, Emmanuel Tiyo Ayikobua, Linda Lillian, Kevin Matama, Shui Ching Nelly Mak, David Onanyang, Theophilus Pius, David Paul Nalumenya, Robinson Ssebuufu, Nina Olivia Rugambwa, Grace Henry Musoke, Kevin Bardosh, Juma John Ochieng, Fred Ssempijja, Patrick Kyamanywa, Gabriel Tumwine, Khalid J. Alzahrani, Susan Christina Welburn

**Affiliations:** 1Infection Medicine, Deanery of Biomedical Sciences, College of Medicine and Veterinary Medicine, The University of Edinburgh, 1 George Square, Edinburgh EH8 9JZ, UK; Ewan.MacLeod@ed.ac.uk (E.T.M.); s.c.n.mak@sms.ed.ac.uk (S.C.N.M.); 2School of Medicine, Kabale University, P.O. Box 317 Kabale, Uganda; hninsiima@kab.ac.ug; 3EcoHealth Alliance, 520 Eighth Ave, Suite 1201, New York, NY 10018, USA; Laudisoit@ecohealthalliance.org; 4School of Health Sciences, Soroti University, P.O. Box 211 Soroti, Uganda or losuwat@sun.ac.ug (L.O.O.); or eayikobua@sun.ac.ug (E.T.A.); 5Department of Pharmacology and Therapeutics, Faculty of Veterinary Medicine, Damanhour University, Damanhour 22511, AlBeheira, Egypt; gaberbatiha@gmail.com; 6Department of Internal Medicine, College of Medicine, Taif University, P.O. Box 11099, Taif 21944, Saudi Arabia; n.edah@tu.edu.sa; 7Kampala International University Western Campus, P.O. Box 71 Bushenyi, Uganda; kyeric007@gmail.com (E.A.); gopama13@gmail.com (I.M.U.); kevicematama@gmail.com (K.M.); piustheophillus@kiu.ac.ug (T.P.); rssebuufu@gmail.com (R.S.); john.juma@kiu.ac.ug (J.J.O.); kalanzifr@yahoo.com (F.S.); pkyamanywa0@gmail.com (P.K.); 8Department of Writing Studies and Rhetoric, Hofstra University, Hempstead, NY 11549, USA; lisa.m.detora@hofstra.edu; 9Faculty of Medicine, Mbarara University of Science and Technology, P.O. Box 1410 Mbarara, Uganda; hnalugo@must.ac.ug; 10Good Clinical Practice Alliance-Europe and Strategic Initiative for Developing Capacity in Ethical Review, BE-1050 Brussels, Belgium; fpc@gcpalliance.org; 11Brigham and Women’s Hospital, Harvard Medical School, Boston, MA 02115, USA; bbierer@bwh.harvard.edu; 12Faculty of Science, Muni University, P.O. Box 725 Arua, Uganda; dc.mwandah@muni.ac.ug; 13College of Veterinary Medicine Animal Resources and Biosecurity, Makerere University, P.O. Box 7062 Kampala, Uganda; katodrago@gmail.com (C.D.K.); nalumenyad@gmail.com (D.P.N.); tumwinegabriel@gmail.com (G.T.); 14Department of Pharmacology and Therapeutics, Faculty of Health sciences, Busitema University, P.O. Box 206 Mbale, Uganda; kiyimbakennedy@gmail.com; 15Uganda National Health Laboratory Services, Ministry of Health, P.O. Box 7272 Kampala, Uganda; lindalilliane32@gmail.com; 16Department of biology, Faculty of Science, Gulu University, P.O. Box 166 Gulu, Uganda; donanyang@yahoo.com; 17Department of Library and Information Science, Faculty of Computing, Library and Information Science, P.O. Box 317 Kabale, Uganda; nrugambwa@kab.ac.ug; 18Faculty of Science and Technology, Cavendish University, P.O. Box 33145 Kampala, Uganda; gracemusoke2medic@gmail.com; 19Center for One Health Research, School of Public Health, University of Washington, Seattle, WA 98195, USA; bardosh_kevin@hotmail.com; 20Department of Clinical Laboratories Sciences, College of Applied Medical Sciences, Taif University, P.O. Box 11099, Taif 21944, Saudi Arabia; Ak.jamaan@tu.edu.sa; 21Zhejiang University-University of Edinburgh Joint Institute, Zhejiang University, International Campus, 718 East Haizhou Road, Haining 314400, China

**Keywords:** COVID-19 clinical trials in resource poor countries, COVID-19, clinical trials in Africa, COVID-19 and medical workers, vaccines, COVAX

## Abstract

Background—misinformation and mistrust often undermines community vaccine uptake, yet information in rural communities, especially of developing countries, is scarce. This study aimed to identify major challenges associated with coronavirus disease 2019 (COVID-19) vaccine clinical trials among healthcare workers and staff in Uganda. Methods—a rapid exploratory survey was conducted over 5 weeks among 260 respondents (66% male) from healthcare centers across the country using an online questionnaire. Twenty-seven questions assessed knowledge, confidence, and trust scores on COVID-19 vaccine clinical trials from participants in 46 districts in Uganda. Results—we found low levels of knowledge (i.e., confusing COVID-19 with Ebola) with males being more informed than females (OR = 1.5, 95% CI: 0.7–3.0), and mistrust associated with policy decisions to promote herbal treatments in Uganda and the rushed international clinical trials, highlighting challenges for the upcoming Oxford–AstraZeneca vaccinations. Knowledge, confidence and trust scores were higher among the least educated (certificate vs. bachelor degree holders). We also found a high level of skepticism and possible community resistance to DNA recombinant vaccines, such as the Oxford–AstraZeneca vaccine. Preference for herbal treatments (38/260; 14.6%, 95% CI: 10.7–19.3) currently being promoted by the Ugandan government raises major policy concerns. High fear and mistrust for COVID-19 vaccine clinical trials was more common among wealthier participants and more affluent regions of the country. Conclusion—our study found that knowledge, confidence, and trust in COVID-19 vaccines was low among healthcare workers in Uganda, especially those with higher wealth and educational status. There is a need to increase transparency and inclusive participation to address these issues before new trials of COVID-19 vaccines are initiated.

## 1. Introduction

Understanding community knowledge and trust has become increasingly important in the design of effective and ethical clinical trials. From 1991 to 2018, Africa contributed only 2.5% to the global total of clinical trials [1]. From a pharmacovigilance standpoint, the continent offers many potential advantages including genetic diversity and a large number of potential participants who are naïve to drug or vaccine products [1]. However, fear, distrust and suspicion are important barriers to trial participation [2,3]. Several factors contribute to skepticism regarding clinical trials and the products they test. Regulations and ethical guidelines to protect patients, while present in Egypt, South Africa, Uganda, and Ghana, are inadequate in many other African countries [1]. Additional factors causing fear and mistrust include a history of inadequate commitment and/or skill on the part of researchers and their staff, shortages of medical personnel, the failure of researchers to understand local culture, poor infrastructure, an absence of national regulatory requirements, and ineffective ethical counseling, community engagement and informed consent processes [1,2,3]. Inadequate human and/or financial resources contribute to the inability to build awareness regarding individual trials [1]. 

Misunderstanding also contributes to widespread myths and fears associated with infectious disease clinical trials. However, it is important to note that such fears are often related, in various ways, to a legacy of distrust due to past medical misconduct and unethical experimentation, which in some cases has led to major international lawsuits [4]. Fear of contracting infectious agents such as the Ebola virus from vaccines (EBOVAC) can also be compounded by psychological trauma following receipt of vaccines [2,3,4,5]. The media, advocacy groups, medical journals, and public information services can each shape how the population receives, analyses, and uses medical and health information. These groups, and social media, have contributed, sometimes inadvertently, to the dissemination of myths and misunderstandings without addressing emotional, psychosocial and ethical aspects of trials [2]. 

To ensure that clinical trials are implemented with the utmost adherence to humane and ethical standards, it is imperative that African research teams increase their human and financial resources, community engagement approaches, training and data collection tools [1,6]. High-quality clinical trials require collaboration with various stakeholders and awareness of the physical, emotional, psychosocial, and ethical needs of potential trial participants and their communities [2]. The emerging consensus is that communities should be inclusively involved in the design, implementation and monitoring and evaluation of such trials to increase trust, acceptability and to negotiate challenges as they arise [7]. Many African countries would benefit from improving their capacity to host clinical trials and investing in research collaborations [8]. A set of common ethical guidelines for the continent as a whole has been suggested as a way to improve both trust and research quality [1]. 

In April 2020, the World Health Organization (WHO), European Union (EU) and France launched the Coronavirus disease 2019 (COVID-19) Vaccines Global Access (COVAX) as a platform to accelerate development and manufacture of COVID-19 vaccines and regulate fair and equitable access of COVID-19 vaccines globally [9]. In December, Moderna and Pfizer/BioNTech vaccines had received ratification in the United States [10]. These mRNA vaccines appear to offer hope to the international community against COVID-19; however, the growth of skepticism continues to undermine vaccine confidence. In addition, increased reliance of local communities on alternative and complementary medicinal options to control infectious diseases furthermore complicates COVID-19 control [11,12]. For example, *Echinacea, Cinchona, Curcuma longa*, and *Curcuma xanthorrhiza* [12] have been recommended; however there is limited information on clinical assessment of their immunogenicity. The objective of the current study was to identify major challenges associated with prospective COVID-19 vaccine clinical trials in Uganda by understanding the perspective of healthcare workers, a group identified as crucial for COVID-19 community management [13]. It was important to assess their knowledge, confidence and trust level on COVID-19 vaccine trials in preparation for the Oxford University–AstraZeneca COVID-19 vaccine program planned to be launched in Uganda in March-April 2021. 

## 2. Materials and Methods

### 2.1. Study Design

A descriptive cross-sectional study was conducted among healthcare workers in health facilities in Uganda from September 5th to October 7th 2020. During this period, COVID-19 national lockdown restrictions were just being lifted, and media reports emphasized the potential benefits of a COVID-19 vaccine. Data were collected using an online questionnaire to minimize printing and contact, consistent with COVID-19 precautionary measures [14,15]. Individuals working in a health facility (clinicians, nurses, pharmacists, laboratory personnel, and support staff) were targeted by using social media (Figure 1). Those who consented to participate in the study were included. Support staff were defined as persons working at the health facility involved in non-administrative activities at the time of the survey. Persons who declined to consent and those not working in a medical facility were excluded. Sample size was determined using the Krejcie and Morgan Table which allows us to determine the sample size of a given finite population at *p* ≤ 0.05 [16]. Since the medical professional population in Uganda was estimated to be 81,982 healthcare workers according to the Uganda Annual health sector performance report 2014/2015 [17], the sample size required was estimated at 390 participants, however 260 participants offered consent after responding to the online survey.

### 2.2. Data Collection and Management

A semi-structured questionnaire was developed after a literature search to identify key areas of concern for community confidence in COVID-19 prevention measures. The questionnaire was divided into four sections—(1) sociodemographic characteristics (age, gender, marital status, educational level, occupation, and location of health facility); (2) knowledge about COVID-19 vaccines and vaccine trials; (3) confidence in vaccine trials, COVID-19 vaccines, the local medical community and government COVID-19 policies; and (4) trust-related questions. The questionnaire was reviewed and validated by 5 different experts in local and international universities with expertise on the topic and then uploaded using a Google form (via docs.google.com/forms (accessed on 9 March 2021)) for pretesting before data collection was conducted. Pretesting was then conducted amongst healthcare workers in selected private healthcare centers in Bushenyi and Mbarara districts (*n* = 30) and these were excluded from the analyzed dataset presented herein. Principal component analysis and further statistical analysis were conducted on the frequencies to check for consistency of responses and a Cranach’s α = 0.8 was considered acceptable.

Questions on knowledge involved binary responses (yes/no) while questions on confidence and trust were ranked using a Likert scale from 0–5, i.e., 0 = very low, 1 = low, 2 = not sure, 3 = moderate, 4 = high, 5 = very high. To reduce guessing responses and bias, the questions on knowledge, confidence and trust were presented randomly to each participant (Appendix A).

The knowledge score was acquired by calculating scoring questions 7–10, 17 and 22. These were then expressed as an average count and converted to percentage and used for analysis. Knowledge questions were on SARS-CoV-2 virology, vaccine development, role of vaccines and research in clinical trials on COVID-19, fear on COVID-19 clinical trials, history of participation in COVID-19 clinical trials (since the government of Uganda is currently conducting preliminary studies), and having received communication on COVID-19 vaccine clinical trials (Appendix A). Our hypothesis was that healthcare workers have a good knowledge on these basic clinical aspects since they have been identified as essential staff and are expected to be vaccinated first ahead of the general population. 

The confidence score was acquired by summing the Likert scores on questions 16, 18–21 and 23, 25, 26 for which the average score was then expressed as a proportion and used for analysis. Questions asked ranged from ranking government commitment to develop a COVID-19 vaccine, ability of Ugandans to handle COVID-19 vaccine clinical trials, commitment of workmates to observe COVID-19 vaccine clinical trials and assess capacity of human resource at the health center to handle COVID-19 vaccine clinical trials.

The trust score was acquired by calculating the average score on questions 11–13, 15, and 24 in which the average score was expressed as a proportion. Questions ranged from the level of fear on COVID-19 vaccine clinical trials, level of suspicion, willingness to participate in COVID-19 clinical trials, and willingness to participate on a rushed COVID-19 vaccine clinical trial (Appendix A).

### 2.3. Statistical Analysis

Data were exported into STATGRAPHICS centurion CVI version 16.1.11 (StatPoint Tech., Inc., Warrenton, VA, USA) and descriptive statistics were conducted after conducting a D’Agostino-pearson omnibus normality test [18] (Appendix A). For narrative description, Likert scores from 0–5 were transformed as follows—0 and 1 were coded for low, 2 and 3 for moderate, while 4 and 5 for high and presented as frequencies. Relationship models for knowledge, confidence, and trust using factorial analysis (FA) and standardized principal component (PC) were conducted followed by multivariable correlation analysis to assess the strength of the relationships. The observed trends in the FA were investigated using General linear Model (GLM) to determine the significant influential variables. All analyses were performed at 95% confidence level and *p*-values less than 0.05 were taken to be significant.

## 3. Results

### 3.1. Population Social Demographic Variables in Ugandan Healthcare Centers

A majority of study participants fell into the middle age category, were men, and had received a college education as shown in Table 1. Most were also either laboratory personnel (31%) or support staff (36%), while 13% were clinicians, 11% nurses and 9% pharmacists. Most preferred vaccines were inactivated vaccines (34.2%), however about 14.6% preferred herbal treatments and organics.

### 3.2. Influence of Sociodemographic Characteristics on the Knowledge Score, Confidence, and Trust for COVID-19 Vaccine Clinical Trials among the Healthcare Workers

Knowledge, confidence, and trust scores for COVID-19 vaccine clinical trials were generally low for all categories. Significant differences in knowledge, trust, and confidence were identified. Of interest, trust scores decreased with increasing education (*p* = 0.001), and confidence and trust levels were higher amongst occupational groups that required less education (Table 2). Confidence and trust levels varied by region, with the highest scores in facilities from eastern Uganda. Participants who preferred the herbal vaccine expressed a relatively higher knowledge on COVID-19 vaccine clinical trials as compared to those who are in favor of a live attenuated vaccine. Trust was also found to be highest in herbal treatments than all the other vaccine types presented in this study (*p* = 0.018).

### 3.3. Descriptive Narrative on Knowledge, Trust and Confidence among Study Participants 

A majority of participants (72.7%, 189/260) falsely stated that Ebola belongs to the coronavirus classification. In addition, a majority of the study participants were unaware of companies involved in COVID-19 vaccine development (65.4%, 170/260), did not think COVID-19 vaccines are necessary to stop the pandemic (87.3%, 227/260), expressed fear towards COVID-19 vaccines (70.8%, 184/260), had no experience on clinical trials (87.7%, 228/260), and had not received any information on the planned COVID-19 vaccine activity in Uganda (79.2%, 206/260). A logistic regression was performed to ascertain the effects of knowledge questions on the likelihood that a participant was a female. The logistic regression model was not statistically significant, χ2(7) = 9.622, *p* = 0.211. The model explained 3.2% (Nagelkerke, *R^2^*) of the variance in the gender and correctly classified 65.8% of cases. Males were more knowledgeable on coronavirus classification (OR = 1.5, 95% CI: 0.7–3.0) and companies involved in COVID-19 vaccine development (OR = 1.8, 95% CI: 1.0–3.2) as compared to their female counterparts (Table 3). 

A majority of study participants who expressed confidence on the COVID-19 clinical trials reported a high workmate commitment to implement COVID-19 control guidelines (140/260, 53.8%), regarded funding as a great challenge to Uganda’s personal investment in COVID-19 vaccines and therapies (145/260, 55.8%), and had a low level of confidence on Ugandan herbal COVID-19 vaccinations (140/260, 53.8%). In addition, a majority of participants (174/260, 66.9%) who expressed distrust for COVID-19 vaccinations for Uganda identified the rushed COVID-19 clinical trials being a major concern (Table 3).

### 3.4. Multivariate Analysis on COVID-19 Clinical Trials amongst Ugandans

From the GLM analysis in Table 4, the sociodemographic factors significantly explained the changes in the confidence (F = 2.74, *p* = 0.001) and trust (F = 5.30, *p* < 0.001), but not knowledge (F = 1.47, *p* = 0.117), with a variability accuracy of 2.65% for knowledge, 9.17% for confidence and 19.92% for trust.

Regression analysis (Table 5) showed that gender was the only significant influential variable (F = 8.49, *p* = 0.0039) for knowledge, while occupation (F = 3.02, *p* = 0.019) and region (F = 6.05, *p* = 0.001) were the significant influential variables for confidence. All sociodemographic variables except age group and gender (*p* > 0.05) were significant contributors to the variation in trust (marital status: F = 5.49, *p* = 0.02; education; F = 3.42; *p* = 0.01; occupation: F = 3.79; *p* = 0.005; region: F = 6.58; *p* < 0.001). 

### 3.5. Relationship between Knowledge, Confidence, and Trust

Eigen analysis of the factor matrix for factorial analysis (FA) produced considerable variations at F2 explaining 84.4% of the cumulative variance in the component, but, with an eigenvalue ≥ 1.0. This left F1 (Eigen value = 1.68) with a cumulative variance of 56.06% as the model component that met the criteria (Table 6).

All three variables—knowledge (55.7%), confidence (81.5%), and trust (84.1%)—were responsible for the variability in factor (component) 1 on a positive multidirectional scale (Table 6), with a closer relationship between confidence and trust (r = 0.527; *p* < 0.001) than knowledge and confidence (r = 0.201; *p* = 0.001) or trust (r = 0.257; *p* < 0.001) (Figure 2).

## 4. Discussion

We identified a low level of knowledge, confidence and trust of COVID-19 vaccine clinical trials amongst healthcare workers in Uganda. In particular, there were no differences in the knowledge, trust and confidence scores with age. These observations highlight mistrust in the community with regard to COVID-19 vaccine clinical trials in Uganda. These findings are in agreement with previous studies in Africa [1,2]. These circumstances signal possible problems for upcoming COVID-19 vaccinations in Uganda. 

The majority of health workers in Uganda believe that the human resources designated to handle COVID-19 cases are inadequate; the health worker evaluation may contribute to antivaccine sentiments, in agreement with previous reports [1]. A previous nationwide study in Uganda showed that healthcare workers are six times more knowledgeable about COVID-19 than teachers (non-medical staff) [19], however a failure to replicate this self-reported knowledge on COVID-19 vaccine clinical trials raises major policy challenges. Our study also identified males as having a significantly higher knowledge score than females, thus identifying gender inequalities that parallel the disproportionate distribution of males and females in the healthcare profession. More educated people were reported to be more reluctant to accept vaccines [20] and this was in agreement with our study.

This is the first study from East Africa demonstrating widespread knowledge and trust gaps related to COVID-19 vaccines amongst healthcare workers. To increase vaccine uptake, community education to improve knowledge on COVID-19 vaccines has been promoted [20,21]. We found that the limited experience in clinical trials among Ugandans also contributes to reluctance and misinformation. This is important since Africa has generally limited experience with major vaccine clinical trials [22]. The Ugandan government’s decision to invest in parallel COVID-19 herbal treatments and therapeutic research through the Ministry of Health (to pursue vaccine research) and Busitema University (to pursue COVID therapies) continues to create further confusion [23,24]. The study also identified mistrust amongst Ugandans towards the herbal treatments, probably arising from the failed Madagascar COVID-Organics cocktail [25,26].

Discrepancies identified in a developing country like Uganda raise major challenges to vaccine distribution goals set by the WHO [9]. Equity education would help promote knowledge among healthcare workers since occupational status has a significant impact on knowledge level, i.e., bachelor’s degree holders are theoretically more knowledgeable than certificate holders [27]. This would also help build confidence in female patients since women talk more freely with other women, leading to more open discussions and helping improve vaccine uptake. The low productivity common in most healthcare centers of Uganda as a result of mistrust [28] only continues to precipitate the low confidence and trust on the planned COVID-19 vaccine clinical trials. This situation would be harmful and unproductive for Ugandans once COVID-19 vaccine uptake is low, since this would undermine the herd immunity offered through vaccination strategies.

Our study has a few limitations. This includes the gender disparities which were consistent with general conditions in the area, including access to education. Women in Uganda are more likely to be employed in nursing and other lower paying positions, leading to under-representation of females in managerial positions [29]. This online questionnaire required a smartphone and internet connectivity, which presented an economic and educational barrier to participation. Globally, there are more females in the healthcare profession than men [30], which suggests an alternate modality should be investigated for future surveys.

We found that the least educated, i.e., illiterate and certificate holders, had a higher confidence and trust level in the COVID-19 vaccine clinical trials than those who had a higher level of education. Findings in the study are in agreement with those from the Democratic Republic of Congo (DRC) in which doctors had a low (27.7%) acceptability for COVID-19 vaccines [31]. In France, healthcare workers were associated with increased vaccine acceptance [32], contrary to findings from Uganda and the DRC (resource-limited countries). The study identified knowledge as a barrier, if not well-nuanced and properly explained in the higher educated people—higher educated people have also more access to internet and hence to the misinformation as well as the real information. Of course, higher education is also associated with the need for greater demand for information about risks and benefits before consent to participate in a trial would be given [33]. This finding may indicate that insufficient information about COVID-19 clinical trials have been given to healthcare workers, and that health professionals do not feel consulted or adequately engaged in trial design and plans. These findings demonstrate challenges for the planned COVID-19 vaccinations in Uganda since medical staff are frontline workers in the global fight against the pandemic [13]. Support staff and nurses were more confident in the COVID-19 vaccine clinical trials than their senior counterparts. The skepticism identified amongst the educated and most professional healthcare workers re-emphasizes the need to increase transparency to encourage scientific and community scrutiny on COVID-19 vaccines [34]. 

Vaccine confidence was lowest in the central and western regions of Uganda and this was important since these are the highly developed regions of Uganda. Our research shows a lack of confidence by the relatively rich and more educated. In addition, the Ugandan herbal treatment is currently marred in a lot of secrecy, however, it is anticipated to be administered orally among COVID-19 patients in Uganda [23,24]. In this study, the largest proportion of Ugandans expressed skepticism against the Live Attenuated Vaccines (LAVs), and DNA Recombinant vaccines (DRVs). The Oxford–AstraZeneca vaccine is a viral vector, i.e., developed from an adenovirus to mimic SARS-CoV-2, thus making it a genetically modified organism and an example of a DRV [35]. Low confidence and trust levels against DRVs identified in this study would raise challenges once Uganda begins to use the Oxford–AstraZeneca vaccines as planned [36]. In addition, the Pfizer/BioNTech vaccines are messenger RNA vaccines—a new class of vaccines [37]—demonstrating a need for more studies in Uganda to promote already trusted and reliable vaccines. Since developing countries lack the capacity to develop vaccines, money spent on the herbal treatments and therapies may be better spent if invested into improved training and funding for basic institutional research to increase transparency and public confidence in scientific reports [1].

This study identifies major challenges to vaccine uptake in Uganda as well as regional differences in opinions. The high fear and mistrust against COVID-19 vaccines identified in this pilot survey were in agreement with previous reports from Liberia and Sierra Leone on EBOVAC studies [2,3]. The skepticism towards COVID-19 vaccines appears to be associated with the fact that vaccine manufacturers and scientists have been predominately from Europe and North America, raising suspicions of neocolonialism through medical research. This shows the need for well-structured clinical trials and drug development to be conducted in resource-poor countries as a strategy to address vaccine hesitancy. 

## 5. Conclusions

Acceptance of COVID-19 vaccines in low-resource countries will probably be stymied by the fact that clinical trials have been conducted outside Africa. To address low trust in COVID-19, but also future pandemic vaccine clinical trials, it is important to situate clinical trials in Africa, led by respected African research institutes, with clear and transparent community engagement, legal and ethical protocols. Future studies should explore community perceptions of mRNA vaccines, since these are the leading vaccine candidates being deployed to control the COVID-19 pandemic despite their transportation logistical cold chain requirement challenge for African countries. Studies should also explore the scientific networks that have emerged around COVID-19 clinical trials and the influence of African researchers, including increasing trust and confidence in vaccines by healthcare workers. Enrolment in the study was stopped after failing to acquire more responses thus a further study involving more study participants would provide more robust data. 

## Figures and Tables

**Figure 1 vaccines-09-00253-f001:**
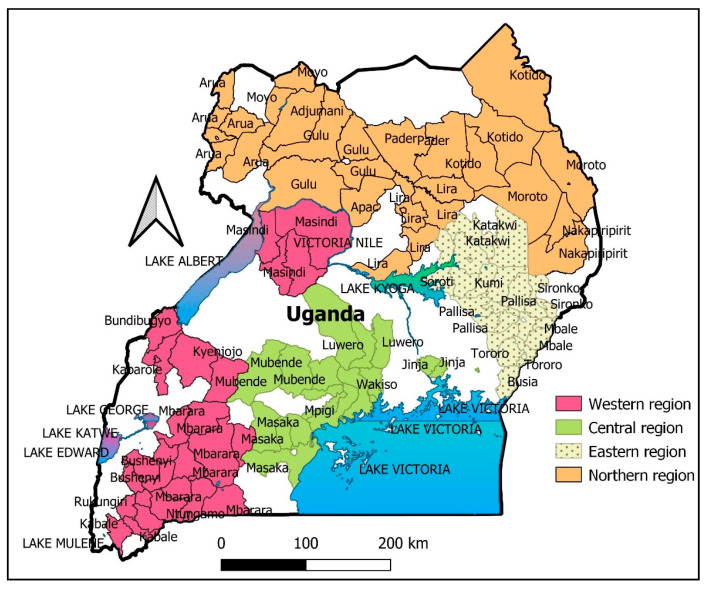
Location of healthcare centers surveyed in the study area by region. In total, participants were based at medical facilities in 46 districts. In particular, 39% of participants came from the central region (101/260), 28% eastern region (72/260), 10% northern region (25/260), and 24% from the western region (62/260). Further individual district demographics indicated more males participated in the study (Appendix A).

**Figure 2 vaccines-09-00253-f002:**
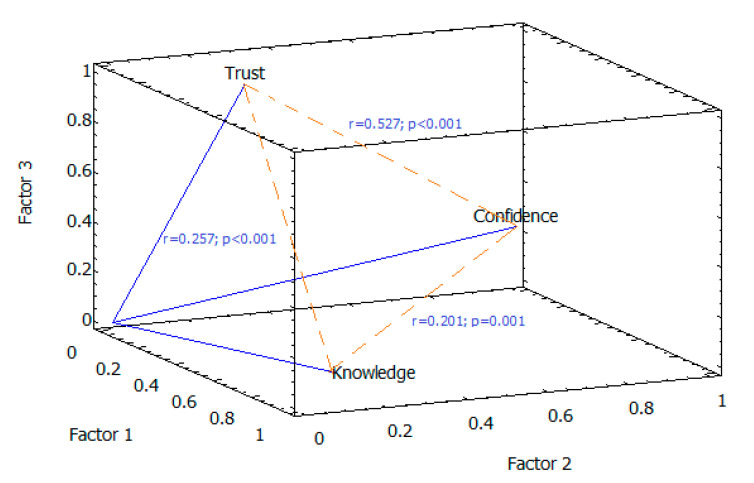
Factor loading components and correlation of variables. The strongest correlation was observed between confidence and trust with factor 2.

**Table 1 vaccines-09-00253-t001:** Statistic on sociodemographic variables in the study population.

Parameter	Variable	Frequency (*n* = 260)	Percent	95% CI
Age (years)	>45	23	8.8	5.8–12.8
	25–45	166	63.8	57.9–69.5
	<25	71	27.3	22.2–33.0
Gender	Female	89	34.2	28.7–40.2
	Male	171	65.8	59.8–71.4
Marital status	Married	118	45.4	39.4–51.5
	Single	142	54.6	48.5–60.6
Education level	Bachelors	107	41.2	35.3–47.2
	Certificate	26	10.0	6.8–14.1
	Diploma	47	18.1	19.8–23.1
	None	8	3.1	1.3–6.0
	Postgraduate	72	27.7	22.5–33.4
Occupation	Clinician	34	13.1	9.4–17.8
	Laboratory personnel	80	30.8	25.4–36.6
	Nurse	29	11.2	7.7–15.4
	Pharmacist	23	8.8	5.8–12.8
	Support staff	94	36.2	30.5–42.1
Location	Central	101	38.8	33.1–44.9
	Eastern	72	27.7	22.5–33.4
	Northern	25	9.6	6.5–13.7
	Western	62	23.8	19.0–29.3
Preferred COVID-19 vaccine type	DNA Recombinant vaccines	41	15.8	11.7–20.6
	Herbal treatments	38	14.6	10.7–19.3
	Inactivated vaccines	89	34.2	28.7–40.2
	Live attenuated vaccines	35	13.5	9.7–18.0
	No vaccine	57	21.9	17.2–27.3
Age (years)	Minimum	18		
	Maximum	65		
	Mean ± SEM	31.8 ± 0.5		

**Table 2 vaccines-09-00253-t002:** Sociodemographic variables associations with knowledge, confidence and trust on COVID-19 vaccine clinical trials in Uganda.

Parameter	Variable	N	Percentage Knowledge Score			Confidence Score			Trust Score		
			Mean ± SEM	ANOVA F (*p*) Value	Median (Min-Max)	Mean ± SEM	ANOVA F(*p*) Value	Median (Min-Max)	Mean ± SEM	ANOVA F(*p*) Value	Median (Min-Max)
Age	>45	23	44.2 ± 3.4	0.603 (0.548)	50 (16.7–66.7)	2.5 ± 0.2	0.2 (0.814)	2.4 (0.9–4.3)	2.4 ± 0.2	1.4 (0.246)	2.6 (0.4–3.8)
	25–45	166	41.4 ± 1.2		33.3 (0.0–83.3)	2.5 ± 0.1		2.5 (0.3–4.6)	2.2 ± 0.1		2.0 (0.0–5.0)
	<25	71	43.4 ± 2.1		33.3 (0.0–83.3)	2.6 ± 0.1		2.7 (0.8–4.4)	2.4 ± 0.1		2.2 (0.4–4.8)
Gender	Female	89	39.3 ± 1.5	4.3 (0.039)	33.3 (0.0–66.7)	2.6 ± 0.1	2.1 (0.147)	2.5 (0.9–4.6)	2.3 ± 0.1	0.1 (0.747)	2.2 (0.4–5.0)
	Male	171	43.7 ± 1.3		50.0 (0.0–83.3)	2.5 ± 0.1		2.5 (0.3–4.4)	2.3 ± 0.1		2.0 (0.0–4.8)
Marital status	Married	118	42.7 ± 1.5	0.2 (0.666)	33.3 (16.7–83.3)	2.6 ± 0.1	0.7 (0.413)	2.5 (0.3–4.4)	2.4 ± 0.1	4.1 (0.045)	2.4 (0.4–5.0)
	Single	142	41.8 ± 1.4		33.3 (0.0–83.3)	2.5 ± 0.1		2.5 (0.8–4.6)	2.2 ± 0.1		2.0 (0.0–4.8)
Education level	Bachelors	107	41.0 ± 1.4	2.3 (0.63)	33.3 (0.0–83.3)	2.5 ± 0.1	1.4 (0.239)	2.5 (0.6–4.1)	2.2 ± 0.1	6.6 (0.001)	2.0 (0.4–4.2)
	Certificate	26	46.2 ± 3.2		50.0 (16.7–66.7)	2.8 ± 0.2		2.8 (1.0–4.6)	2.9 ± 0.3		3.0 (0.4–5.0)
	Diploma	47	46.1 ± 2.6		50.0 (16.7–83.3)	2.5 ± 0.1		2.5 (1.0–4.4)	2.4 ± 0.1		2.0 (0.6–4.8)
	None	8	31.3 ± 6.6		33.3 (0.0–50)	3.1 ± 0.4		3.1 (1.6–3.9)	3.1 ± 0.4		3.0 (2.0–4.8)
	Postgraduate	72	41.2 ± 1.9		33.3 (16.7–83.3)	2.5 ± 0.1		2.5 (0.3–4.6)	2.0 ± 0.1		2.0 (0.0–3.4)
Occupation	Clinician	34	39.7 ± 2.4	0.4 (0.825)	33.3 (16.7–83.3)	2.1 ± 0.1	3.5 (0.009)	2.2 (0.3–3.6)	2.1 ± 0.1	6.6 (0.001)	2.0 (0.6–3.6)
	Laboratory personnel	80	42.5 ± 1.8		50.0 (0.0–83.3)	2.5 ± 0.1		2.5 (0.8–4.3)	1.9 ± 0.1		2.0 (0.0–4.4)
	Nurse	29	42.5 ± 3.3		50.0 (16.7–83.3)	2.5 ± 0.1		2.5 (1.0–4.1)	2.5 ± 0.1		2.5 (1.2–5.0)
	Pharmacist	23	44.9 ± 3.2		50.0 (16.7–83.3)	2.4 ± 0.2		2.3 (0.9–4.6)	2.2 ± 0.2		2.0 (1.0–4.0)
	Support staff	94	42.0 ± 1.8		33.3 (0.0–83.3	2.7 ± 0.1		2.7 (0.6–4.6)	2.6 ± 0.1		2.4 (0.4–4.8)
Location	Central	101	42.4 ± 1.5	1.8 (0.144)	50 (16.7–83.3)	2.5 ± 0.1	6.4 (0.001)	2.5 (0.9–4.6)	2.1 ± 0.1	10.8 (0.001)	2.0 (0.4–4.4)
	Eastern	72	44.9 ± 1.9		50 (0.0–83.3)	2.8 ± 0.1		2.9 (0.8–4.4)	2.8 ± 0.1		2.8 (0.6–5.0)
	Northern	25	36.7 ± 3.2		33.3 (16.7–83.3)	2.2 ± 0.2		2.1 (0.9–3.4)	2.4 ± 0.2		2.2 (1.2–3.8)
	Western	62	40.9 ± 2.3		33.3 (0.0–83.3)	2.3 ± 0.1		2.4 (0.3–4.1)	2.0 ± 0.1		2.0 (0.0–3.8)
Preferred COVID-19 vaccine	DRV	41	42.3 ± 2.9	1.2 (0.319)	33.3 (16.7–83.3)	2.5 ± 0.1	1.0 (0.412)	2.5 (0.6–4.1)	2.2 ± 0.1	3.1 (0.018)	2.0 (0.8–4.0)
	HV	38	46.9 ± 2.8		50 (16.6–83.3)	2.7 ± 0.1		2.6 (1.3–4.3)	2.5 ± 0.2		2.6 (0.8–4.6)
	IV	89	41.0 ± 1.5		33.3 (0.0–83.3)	2.5 ± 0.1		2.5 (0.8–4.6)	2.1 ± 0.1		2.0 (0.0–4.2)
	LAV	35	39.5 ± 2.5		33.3 (16.7–83.3)	2.5 ± 0.1		2.6 (0.9–4.3)	2.3 ± 0.2		2.2 (0.4–4.4)
	None	57	42.2 ± 1.0		33.3 (0.0–83.3)	2.4 ± 0.1		2.4 (0.3–4.6)	2.6 ± 0.2		2.4 (0.6–5.0)

KEY: DRV = DNA Recombinant vaccines, HV = Herbal treatments, IV = Inactivated vaccines, LAV = Live attenuated vaccines. N = number of participants, SEM = Standard error mean, Min-Max = Minimum-Maximum values, ANOVA = Analysis of variance, P-probability value.

**Table 3 vaccines-09-00253-t003:** Descriptive narrative on knowledge, trust and confidence among study participants.

Variables	Variable	Proportions by Gender			*p* Value	OR (95% CI)
		Females	Males	Total		
Beta coronaviruses include the following except	Ebola	67 (25.8)	122 (46.9)	189 (72.7)		1
	MERS	9 (3.5)	17 (6.5)	26 (10.0)	0.684	1.2 (0.5–2.9)
	SARS	13 (5.0)	32 (12.3)	45 (17.3)	0.316	1.5 (0.7–3.0)

Do you know any company involved in COVID-19 vaccine development?	No	65 (25.0)	105 (40.4)	170 (65.4)	0.047	1
	Yes	24 (9.2)	66 (25.4)	90 (34.6)		1.8 (1.0–3.2)

Do you think breaking the COVID-19 circle involves vaccine development?	No	12 (4.6)	21 (8.1)	33 (12.7)	0.685	1
	Yes	77 (29.6)	150 (57.7)	227 (87.3)		1.2 (0.5–2.6)

Do you have fear about the COVID-19 vaccine?	No	28 (10.8)	48 (18.5)	76 (29.2)	0.310	1
	Yes	61 (23.5)	123 (47.3)	184 (70.8)		1.3 (0.8–2.4)

Have you ever participated in any clinical trial previously?	No	79 (30.4)	149 (57.3)	228 (87.7)	0.949	1
	Yes	10 (3.8)	22 (8.5)	32 (12.3)		1.0 (0.5–2.3)

I have received adequate communication on the COVID-19 vaccine trials in Uganda	No	73 (28.1)	133 (51.2)	206 (79.2)	0.569	1
	Yes	16 (6.2)	38 (14.6)	54 (20.8)		1.2 (0.6–2.4)
Variables				Frequencies on participants responses on COVID-19		
				High	Low	Moderate
Confidence on COVID-19 vaccinations						
I have been enlightened on WHO guidelines and stages for vaccine trials				47 (18.1)	120 (46.2)	93 (35.8)
Rank the Ugandan government’s commitment to the development of a genuine COVID-19 vaccine and therapy?				54 (20.8)	101 (38.8)	105 (40.4)
Confidence in the skills of Ugandans and their ability to handle the COVID-19 clinical trial?				98 (37.7)	44 (16.9)	118 (45.4)
My workmates’ committment to COVID-19 control guidelines				140 (53.8)	18 (6.9)	102 (39.2)
There are sufficient designated medical personnel handling COVID-19 cases at my workplace?				93 (35.8)	57 (21.9)	110 (42.3)
My information about the planned COVID-19 vaccinations in Uganda				41 (15.8)	136 (52.3)	83 (31.9)
Level of challenge posed by access to funding in vaccine development for Ugandan COVID-19 vaccine clinical trials				145 (55.8)	21 (8.1)	94 (36.2)
My level of confidence in herbal COVID-19 treatments being promoted in Uganda				51 (19.6)	140 (53.8)	69 (26.5)
Trust on COVID-19 vaccinations						
Level of fear				104 (40.0)	69 (26.5)	87 (33.5)
Level of suspicion				118 (45.)	58 (22.3)	84 (32.3)
Willingness to participate in COVID-19 clinical trials				60 (23.1)	123 (47.3)	77 (29.6)
Willingness to participate in a COVID-19 clinical trial				41 (15.8)	174 (66.9)	45 (17.3)
Level of trust for the Ugandan national regulatory guidelines for clinical trials				57 (21.9)	86 (33.1)	117 (45.0)

KEY: OR = Odds ratios, 95% CI = confidence intervals.

**Table 4 vaccines-09-00253-t004:** Variable influence of the knowledge score, confidence, and trust.

Source	SS	Df	MS	F-Ratio	*p*-Value	R-sq	R-sq (adj)
**Knowledge**							
Model	5589.81	15	372.654	1.47	0.117	8.29	2.65
Residual	61845.2	244	253.464				
Total (Corr.)	67435	259					
**Confidence**							
Model	24.8141	15	1.654	2.74	0.001	14.43	9.17
Residual	147.151	244	0.603				
Total (Corr.)	171.965	259					
**Trust**							
Model	59.5571	15	3.970	5.30	<0.001	24.56	19.92
Residual	182.937	244	0.750				
Total (Corr.)	242.494	259					

Note: Corr., Corrected; SS, Sum of Squares; MS, Mean Square; DF, Degree of freedom; R-sq., Correlation squared (accuracy); adj., adjusted.

**Table 5 vaccines-09-00253-t005:** Regression model outcome summary and significance of predictor variables.

Source	SS	Df	MS	F-Ratio	*p*-Value	Variance
**Knowledge**						
Age group	313.274	2	156.637	0.62	0.5399	−2.105
Gender	2152.75	1	2152.750	8.49	0.0039	21.888
Marital Status	68.2873	1	68.287	0.27	0.6042	−2.205
Education	1336.58	4	334.144	1.32	0.2637	1.948
Occupation	626.993	4	156.748	0.62	0.6498	−2.392
Region	1054.89	3	351.630	1.39	0.2472	1.857
Residual	61845.2	244	253.464			253.464
Total (corrected)	67435	259				
**Confidence**						
Age group	0.43659	2	0.218	0.36	0.697	−0.008
Gender	0.39342	1	0.393	0.65	0.420	−0.002
Marital Status	1.67448	1	1.674	2.78	0.097	0.013
Education	2.91125	4	0.728	1.21	0.309	0.003
Occupation	7.28699	4	1.822	3.02	0.019	0.030
Region	10.944	3	3.648	6.05	0.001	0.058
Residual	147.151	244	0.603			0.058
Total (corrected)	171.965	259				
**Trust**						
Age group	2.44901	2	1.225	1.63	0.197	0.010
Gender	2.64235	1	2.642	3.52	0.062	0.022
Marital Status	4.11783	1	4.118	5.49	0.020	0.040
Education	10.252	4	2.563	3.42	0.010	0.044
Occupation	11.3622	4	2.841	3.79	0.005	0.052
Region	14.7894	3	4.930	6.58	0.000	0.079
Residual	182.937	244	0.750			0.750
Total (corrected)	242.494	259				

Note: Corr., Corrected; SS, Sum of Squares; MS, Mean Square; DF, Degree of freedom; R-sq., Correlation squared (accuracy); adj., adjusted.

**Table 6 vaccines-09-00253-t006:** Descriptive characteristics and factor loading matrix of variables in factorial analysis (FA).

Variables	Average ± SD	Factor 1 ^α^	Factor 2 ^α^	Estimated Communality	Specific Variance
Knowledge	42.18 ± 16.14	0.557	−0.828	0.636	0.364
Confidence	2.52 ± 0.81	0.815	0.341	0.674	0.326
Trust	2.29 ± 0.97	0.841	0.218	0.547	0.453

Note: F1 [Eigen value, 1.68169; cumulative %, 56.056], F2 [Eigen value, 0.849384; cumulative %, 84.369], ^α^ [factor loading matrix before rotation].

## Data Availability

The data presented in this study are openly available in figshare at https://figshare.com/s/9f89827c45921ec5196a (accessed on 9 March 2021).

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
