# Peer review of "A Descriptive-Multivariate Analysis of Community Knowledge, Confidence, and Trust in COVID-19 Clinical Trials among Healthcare Workers in Uganda"

_vaccines, 2021, doi:10.3390/vaccines9030253_

Round 1

Reviewer 1 Report

I was invited to review the paper entitled "A descriptive-multivariate analysis of community knowledge, confidence, and trust in COVID-19 clinical trials among healthcare workers in Uganda". It was a cross sectional study aimed to investigate the confidence of healthcare workers about COVID-19 trials in Uganda. The topic in my opinion is relevant and it is the first paper about it from African countries. Despite this, I have some relevant concerns:

  • Sample size estimation was not performed;
  • validation procedure of the questionnaire is missing;
  • Support staff was not defined. Probably they were not healthcare workers;
  • Discussion section is poor and should be improved. Also, comparison with similar studies was missing.

Minor comments:

  • Continous variables were not checked for normal distributrion;
  • THe department of work should be considered as covariate

Author Response

Comment 1: Sample size estimation was not performed

Response 1: Sample size was determined using the Krejcie and Morgan Table since the medical professionals population in Uganda was estimated to be 81 982 healthcare workers. This implied we required about 390 participants, however only 260 participants offered consent after responding to the online survey.

We have now included this clarifying information in the manuscript (lines 148-152).

Comment 2: Validation of questionnaire.

Response 2: The questionnaire was validated by first sharing it with international experts in the field of global health. Pretesting was then conducted amongst healthcare workers in selected private healthcare centers in Bushenyi and Mbarara districts (n=30) and these were excluded from the data collected. Principal component analysis and analysis was conduct-ed on the questions to check for consistency of responses and a Cronbach’s alpha of 0.8 was considered acceptable.

We have added this information to the manuscript (lines 171-175).

Comment 3: Support staff was not defined. Probably they were not healthcare workers;

Response 3: Support staff were defined as persons working at the health facility involved in non-administrative activities at the time of the survey. The definition of support staff is included in the manuscript on lines 145-147.

  • Comment 4: Discussion section is poor and should be improved. Also, comparison with similar studies was missing.

Response 4: This is the first study from East Africa on the subject. Essentially all studies of vaccine hesitancy, community confidence, and trust have been performed in developed countries and thus were not appropriate to compare with situations in developing countries.

Minor comments:

  • Comment 5: Continuous variables were not checked for normal distributrion;

Response 5: This was checked however this was a preliminary process and not part of the results. A note has been added under statistical analysis, however. We have included present both the mean and median to provide more detailed information.

Reviewer 2 Report

Dear Authors, 

For the first time in my scientific work I have come across a manuscript to which I have no substantive or linguistic remarks. Searching for minor errors "by force" is not the role of a reviewer, so I have no critical remarks.

Very well written article. I recommend that it be accepted for publication in its current form.

Author Response

Thank you very much for the encouraging comments. We look forward to making contributions to the vaccine effort (GAVI) to support scientists in developing countries.

Reviewer 3 Report

Overall this is a great paper and well written. I only have minor comments:

You mention "herbal vaccines" throughout, but would a better term be "herbal treatment"? Even "herbal prophylactic" might be preferable to using the word "vaccine".

For Table 3, I was a bit confused by "Frequencies on gender" - I think "Proportion by gender" or "Proportion by sex" is better, and should it be Female and Male (not Sex). I would also present the column % (e.g., for Female and Ebola it would be 67/89). And could you footnote what the P-value means in this table? I'm also a bit confused as to what the OR means in this table. If it is not a multivariable analysis, then the chi-sq p-value is probably sufficient.

I would also mention how uptake of COVID-19 vaccine in LMICs will probably be stymied by limited clinical trials there.

Author Response

Comment 1: Overall this is a great paper and well written. I only have minor comments

Response 1: Thank you very much. We hope the paper makes a significant contribution to vaccination distribution, and most particularly the AstraZeneca product, that will be rolled out in the coming months in Uganda.

Comment 2: You mention "herbal vaccines" throughout, but would a better term be "herbal treatment"? Even "herbal prophylactic" might be preferable to using the word "vaccine".

Response 2: Edited as advised.

Comment 3: For Table 3, I was a bit confused by "Frequencies on gender"

Response 3: Edited to remove ambiguity.

Comment 4: P-value means in this table? I'm also a bit confused as to what the OR means in this table. If it is not a multivariable analysis, then the chi-sq p-value is probably sufficient.

Response 4: A key has been provided.A logistic regression analysis was conducted through which the p values were generated. True X2 would be sufficient, however 95% CI are provided to provide more information to help the reader to interpret the findings with caution.

Comment 5: I would also mention how uptake of COVID-19 vaccine in LMICs will probably be stymied by limited clinical trials there.

Response 5: Edited as advised (conclusion).

Reviewer 4 Report

The authors provide a survey on knowledge, trust and confidence of healthcare professionals (structured by status) about COVID-19 vaccines in Uganda. I am not quite sure this kind of studies satisfies the scope of a journal such as Vaccines but, if so, the authors have gained quite informative insights. This information may be important to achieve local herd immunity against COVID-19 and eventually at a worldwide level.

Due to the particular features of many African countries, measures to reinforce the global opinion on formal prophylaxis is central to make this goal. Sociodemographic aspects, cultural concerns, viral infection knowledge, low R&D tradition, unbalances between male-female opportunities and even post-colonialism issues are discussed. Taking these answers into consideration, the authors have identified bottlenecks for prophylactic success and provide a solution consisting on strong awareness campaigns mainly directed to the healthcare staff. Such campaigns should communicate that current, molecular biology-based,  vaccine formulations show several advantages over those based on inactivated microorganisms, and both definitely over herbal remedies.

Finally, I find subject sampling, variables and statistical methods are appropriate to address the proposed challenges. The manuscript is very well-written and authoritative, and show strong knowledge about the local idiosyncrasy pertaining vaccination. However, I still have a couple of concerns that I think should be addressed.

Major points

1) Beyond authors have definitely identified obstacles for COVID-19 prophylaxis in Uganda, sociodemographic status appears more important than the vaccine knowledge itself. Knowledge can be somehow improved by campaigns and inter-female confident speaking. However, do the authors know how to approach the sociodemographic bottleneck or is this an dead-ended road in the country? Please discuss.

2) Authors claim that, ironically, the better educated people is more reluctant to formal vaccination than the rest. A higher access to the internet have been raised as a possible reason to explain this fact. This is probably true but somehow counterintuitive since it implies that fake information is more powerful persuading the users than the right directions of the network. Predictably, the same will also happen for the remaining people as they gain global access to the internet, which are not immune to desinformation either. The question here is, should also strong prosecution measurements against misleading disseminators be taken? I want to point this since such spreaders would be ultimately observed as guilty of the death of many people. Please discuss.

Minor points:

1) There appears to be some problems with Table 2 (rightmost section broken) and Table 3 (should not “Sex” be substituted by “Male” in the header? ).

Author Response

Comment 1: The authors provide a survey on knowledge, trust and confidence of healthcare professionals (structured by status) about COVID-19 vaccines in Uganda. I am not quite sure this kind of studies satisfies the scope of a journal such as Vaccines but, if so, the authors have gained quite informative insights. This information may be important to achieve local herd immunity against COVID-19 and eventually at a worldwide level.

Response 1: Thank you very much. We believe the study will make a contribution to help reverse the lack of commitment in the general population, although the need for more studies in Africa cannot be ignored.

Comment 2: Due to the particular features of many African countries, measures to reinforce the global opinion on formal prophylaxis is central to make this goal. Sociodemographic aspects, cultural concerns, viral infection knowledge, low R&D tradition, unbalances between male-female opportunities and even post-colonialism issues are discussed. Taking these answers into consideration, the authors have identified bottlenecks for prophylactic success and provide a solution consisting on strong awareness campaigns mainly directed to the healthcare staff. Such campaigns should communicate that current, molecular biology-based,  vaccine formulations show several advantages over those based on inactivated microorganisms, and both definitely over herbal remedies.

Response 2: Thank you very much. Authors remain in agreement with this recommendation. Finally, I find subject sampling, variables and statistical methods are appropriate to address the proposed challenges. The manuscript is very well-written and authoritative, and show strong knowledge about the local idiosyncrasy pertaining vaccination. However, I still have a couple of concerns that I think should be addressed. Thank you very much.

Minor points:

Comment 3: 1) There appears to be some problems with Table 2 (rightmost section broken) and Table 3 (should not “Sex” be substituted by “Male” in the header?).

Response 3: This was a formatting problem. The page has been changed to landscape.

Round 2

Reviewer 1 Report

I revised the second version of the paper "A descriptive-multivariate analysis of community knowledge, confidence, and trust in COVID-19 clinical trials among healthcare workers in Uganda". Despite previous suggestions, Authors did not addressed all comment.

Comment 1: Why did Authors stopped the enrollment despite the minimum sample size was not reached? In my opinion this calculation was not performed before the paper was wrote. In addition, I'm not in accord with Authors with the methods performed to calculate it.

Comment 2: Results of validation for each section should be reported at least a supplementary material. 

Comment 3: Ok.

Comments 4: Readers of this journal are from all parts of the worlds so are interested in comparison with study from other countries. Please add it in discussion sections.

Comment 5: Ok.

Author Response

Author response to Reviewer 1 comments on Manuscript ID: vaccines-1100575 Round Two

Comments and Suggestions for Authors

I revised the second version of the paper "A descriptive-multivariate analysis of community knowledge, confidence, and trust in COVID-19 clinical trials among healthcare workers in Uganda". Despite previous suggestions, Authors did not addressed all comment.

Comment 1: Why did Authors stopped the enrollment despite the minimum sample size was not reached? In my opinion this calculation was not performed before the paper was wrote. In addition, I'm not in accord with Authors with the methods performed to calculate it.

Response 2: The Ugandan government announced plans to conduct covid19 clinical trials in partnership with the UK University on July 10, 2020. These clinical trials were planned to kick off in November 2020 however, these didn’t materialize [1]. On September 16, 2020, the Ugandan Ministry of Health commenced clinical trials on the use of convalescent plasma therapy [2]. Basing on our knowledge regarding government discussions on the UK Oxford vaccine planned for November 2020, we initiated this study in late August 2020. During this period, we had already conducted a couple of preliminary studies on covid19 and realized the need to quantify challenges to the planned covid19 vaccine trials which were pushed forward for 2021 since WHO had communicated that developing countries should wait for late 2021/22 to expect the vaccine. In addition, the fact that developing countries didn’t participate in funding for covid19 vaccine development meant priority would be given to developed nations to access the vaccine [3], as is currently the case. Had Uganda succeeded to participate in the initial clinical trials, probably, this would have lowered the current price of each covid19 vaccine which is being bought at USD 7 (for 18 million doses) compared to Brazil, Saudi Arabia and South Africa which participated and are paying $5.25 per dose [4]. On January 27, 2021, the Ugandan president officially launched the herbal treatments initiatives to treat covid19 under the Ministry of Health and Busitema University and a parallel project to procure vaccines from AstraZeneca was established [5]. The survey dates on the data collected showed that respondents had stopped responding to the questionnaire by Oct 3, 2020. We waited for an extra three months and in January 2021, the survey was closed after failing to attract more participants from the healthcare centers. This forced our sample size to reduce and wouldn’t wait any further since Uganda is planning to implement covid19 vaccinations in March. Findings in the study are important to the government since this will influence policy to avoid pitfalls associated with vaccine hesitance.

 [1]. The Independent. https://www.independent.co.ug/uganda-in-talks-with-uk-university-to-take-part-in-covid-19-vaccine-trials/ Published by the Independent on July 10, 2020. Accessed Feb 19, 2021.

[2]. Ministry of Health Uganda. Uganda advances clinical trials for treatment for covid19. https://www.independent.co.ug/uganda-in-talks-with-uk-university-to-take-part-in-covid-19-vaccine-trials/ Published by Ministry of Health Uganda, September 16, 2020. Accessed Feb 19, 2020.

[3]. The Guardian. Most poor nations 'will take until 2024 to achieve mass Covid-19 immunisation' published by the Guardian on Jan 27, 2021. URL: https://www.theguardian.com/society/2021/jan/27/most-poor-nations-will-take-until-2024-to-achieve-mass-covid-19-immunisation. Accessed Feb 09, 2021.

[4]. Reuters. UPDATE 1-Uganda orders 18 mln doses of AstraZeneca's COVID-19 vaccine, gov't says. Published on Feb 3, 2021. URL: https://www.reuters.com/article/health-coronavirus-uganda-idUSL1N2K91VC Accessed Feb 19, 2021.

[5]. Nobert Atukunda. Scientists upbeat about Uganda’s Covid-19 drug. Published by Daily Monitor on Jan 28, 2021. URL: https://www.monitor.co.ug/uganda/news/national/scientists-upbeat-about-uganda-s-covid-19-drug-3271862  Accessed Feb 19, 2021

Response 2: The Ugandan government announced plans to use the Oxford vaccine in

Response 2: in March 2021.

Comment 2: Results of validation for each section should be reported at least a supplementary material. 

Response 2: D'Agostino-P omnibus test normality results are provided as supplementary file 3.

Comment 3: Ok.

Response 3: Thank you.

Comments 4: Readers of this journal are from all parts of the worlds so are interested in comparison with study from other countries. Please add it in discussion sections.

Response 4: Edits as shown below (see page 14).

Findings in the study are in agreement with those from the Democratic Republic of Congo (DRC) in which doctors had a low (27.7%) acceptability for COVID-19 vaccines [31]. In France, healthcare workers were associated with increased vaccine acceptance [32], contrary to findings from Uganda and the DRC (resource limited countries).

Comment 5: Ok.

Response 5: Thank you.

Round 3

Reviewer 1 Report

About first comment, as reported by Authors, Ugandan government announced plans to conduct covid19 clinical trials but this paper is not a trial report. So I do not understand the reason of stopping enrollment. If the Editor is in accord with these reasons, Authors have to report them on methods and discuss it among limitation of this study.

About comment 2, I required reporting results of validation procedure and not normality tests.

Author Response

Comment 1: About first comment, as reported by Authors, Ugandan government announced plans to conduct covid19 clinical trials but this paper is not a trial report. So I do not understand the reason of stopping enrollment. If the Editor is in accord with these reasons, Authors have to report them on methods and discuss it among limitation of this study.

Response 1: This has been reported as a limitation under the conclusion:

Enrolment in the study was stopped after failing to acquire more responses thus a further study involving more study participants would provide more robust data.

Comment 2: About comment 2, I required reporting results of validation procedure and not normality tests.

Response 2: Validity was conducted however these were not relevant for the body of results. Raw data files have been shared through Figshare https://figshare.com/s/9f89827c45921ec5196a for readers to interpret findings in this study with caution.